# Molecular Cloning, Characterization, and Application of a Novel Multifunctional Isoamylase (MIsA) from *Myxococcus* sp. Strain V11

**DOI:** 10.3390/foods13213481

**Published:** 2024-10-30

**Authors:** Siting Feng, Weiqi Zhang, Jun Liu, Yusen Hu, Jialei Wu, Guorong Ni, Fei Wang

**Affiliations:** 1College of Bioscience and Bioengineering, Jiangxi Agricultural University, Nanchang 330045, China; fst15797897830@163.com (S.F.); zwq13707923026@163.com (W.Z.); 15007091566@163.com (J.L.); hys15572599217@163.com (Y.H.); 18861748005@163.com (J.W.); 2College of Land Resources and Environment, Jiangxi Agricultural University, Nanchang 330045, China; 3Institute of Ecological Restoration Innovation of Zhongke Jiangxi, Nanchang 330045, China

**Keywords:** multifunctional starch debranching enzyme, *Myxococcus* sp. strain V11, isoamylase, 4-alpha-glucanotransferase, maltotetraose

## Abstract

A novel multifunctional isoamylase, MIsA from *Myxococcus* sp. strain V11, was expressed in *Escherichia coli* BL21(DE3). Sequence alignment revealed that MIsA is a typical isoamylase that belongs to glycoside hydrolase family 13 (GH 13). MIsA can hydrolyze the α-1,6-branches of amylopectin and pullulan, as well as the α-1,4-glucosidic bond in amylose. Additionally, MIsA demonstrates 4-α-D-glucan transferase activity, enabling the transfer of α-1,4-glucan oligosaccharides between molecules, particularly with linear maltooligosaccharides. The *K*_m_*, K*_cat_, and *V*_max_ values of the MIsA for amylopectin were 1.22 mM, 40.42 µmol·min^–1^·mg^–1^, and 4046.31 mM·min^–1^. The yields of amylopectin and amylose hydrolyzed into oligosaccharides were 10.16% and 11.70%, respectively. The hydrolysis efficiencies were 55%, 35%, and 30% for amylopectin, soluble starch, and amylose, respectively. In the composite enzyme hydrolysis of amylose, the yield of maltotetraose increased by 1.81-fold and 2.73-fold compared with that of MIsA and MTHase (MCK8499120) alone, respectively.

## 1. Introduction

Starch, a carbohydrate composed of glucose units, is abundant in nature. It functions as a primary energy source for humans and animals and serves as a critical raw material in various industries, including food, chemicals, and pharmaceuticals [1]. Starch is composed of amylose (15% to 25% of starch content), characterized by linear glucose units linked by α-1,4-glycosidic bonds, and amylopectin (75% to 85% of starch content), featuring multiple oligosaccharide branches attached to the main chain through α-1,6-glycosidic bonds [2]. The hydrolysis of α-1,4-glycosidic bonds within starch molecules is efficiently catalyzed by many starch-processing enzymes; however, their ability to cleave α-1,6-glycosidic bonds is relatively limited. This constraint results in reduced starch conversion rates and compromised product quality. Starch debranching enzymes (SDBEs) specifically hydrolyze α-1,6-glycosidic bonds in starch molecules. The combination of SDBEs with enzymes that hydrolyze α-1,4-glycosidic bonds enables complete starch hydrolysis in starch processing industries, thereby enhancing starch conversion rates [3].

Based on their catalytic mechanisms and substrate specificities, starch debranching enzymes are classified into pullulanase (EC. 3.2.1.41, Type I pullulanase), amylopullulanase (EC. 3.2.1.41, Type II pullulanase), and isoamylase (EC. 3.2.1.68) [3]. Pullulanase features a typical (β/α)_8_-barrel structure and belongs to the glycoside hydrolase GH13 family. It efficiently hydrolyzes pullulan, with the minimal catalytic unit being maltotriose-α-1,6-maltotriose, but it has lower activity toward high-molecular-weight branched starch [4]. In the carbohydrate active enzymes (CAZy) database, pullulanases are classified under the glycoside hydrolase families GH13 [5] and GH57 [6], which possess a typical (β/α)_7_-barrel structure. Pullulanase acts as a bifunctional debranching enzyme, catalyzing both α-1,4-glucosidic and α-1,6-glucosidic bond hydrolysis, resulting in greater activity toward pullulan than toward branched starch [7,8]. Amylopullulanase, which is also a member of the GH13 family of glycoside hydrolases, shows high hydrolytic activity toward branched starch and glycogen but lower activity toward low-molecular-weight cyclodextrins. The minimal catalytic unit is maltotriose-α-1,6-maltotetraose, which does not hydrolyze pullulan [9,10,11].

Isoamylases, also belonging to the glycoside hydrolase GH13 family, show high hydrolytic activity toward amylopectin and glycogen and lower hydrolytic activity toward low-molecular-weight dextrins, with the smallest unit of action being maltotriosyl-α-1,6-maltotetrasaccharide, and are unable to hydrolyze pullulan [6,12,13].

The gene for an isoamylase from *Pseudomonas amyloderamosa* was first cloned and expressed by Amemura et al. in 1988 [14], leading to industrial production under the trade name Promo-zyme^®^D2. However, Promo-zyme^®^D2 exhibited a specific enzyme activity of only 180 U mg^–1^ [10,14]. In contrast, the isoamylase IsoM from *Corallococcus* sp. EGB demonstrated a significantly higher specific enzyme activity of 70,600 U mg^–1^, as determined by the iodine staining method [15]. Nevertheless, IsoM expression in yeast remains challenging, with low levels achieved, and industrial-scale production has yet to be realized. Domestically developed isoamylases with independent intellectual property rights are currently hindered by limitations such as low enzyme activity, instability, and inadequate expression levels. Consequently, commercial products available in the market primarily comprise imported enzyme preparations.

Recently, starch debranching enzymes that exhibit high activity or distinct biochemical properties have garnered attention due to their significant potential applications in the production of sugar syrup, resistant starch, and cyclodextrin. The synergistic catalysis between SDBEs and other starch-active hydrolases can significantly enhance raw material utilization and production efficiency during key starch processing steps, such as saccharification and modification [4].

In the preliminary stages of this project, a strain of *Myxococcus* sp. V11 was isolated [16]. Subsequent genome sequencing revealed the presence of a coding sequence (MCK8502360.1) for a starch debranching enzyme (MIsA). In this study, MIsA was heterologously expressed and characterized. A novel multifunctional isoamylase with amylose-hydrolyzing activity and glycosyltransferase activity has been reported. The combination of MIsA with other α-amylase facilitates the production of a greater quantity of oligosaccharides.

## 2. Materials and Methods

### 2.1. Bacterial Strains, Culture Conditions, and Plasmids

*The Myxococcus* sp. strain V11 was cultivated in VY/2 media [16]. *Escherichia coli* DH5α (TransGen Biotech, Beijing, China) was used to construct a recombinant plasmid for starch debranching enzymes. Gene cloning and DNA sequencing were performed using a one-step cloning kit (Vazyme Biotech, Nanjing, China), and the pET-29a (+) plasmid was used as an expression vector to express starch debranching enzymes in *E. coli* BL21(DE3) (TransGen Biotech, Beijing, China).

### 2.2. Gene Cloning, Sequencing, and Construction of the Expression Vector

Based on the DNA sequence of the annotated hypothetical isoamylase, the gene encoding isoamylase was amplified via PCR using the primer pair *MIsA* F (5′-GGTTCCATGGCTGATATCGGATCCATGAGGAGGGCCGAG-3′) and *MIsA* R (5′-GTGCTCGAGTGCGGCCGCAAGCTTCTCC GTCGACGGCCG-3′). The PCR-amplified fragment was subsequently sequenced by Tsingke Corporation (Hunan, China).

The PCR products were digested using *Bam H* I and *Hin d* III and inserted into the *Bam H* I–*Hin d* III sites of pET-29a (+) to obtain the plasmid pET-29a (+)-*MIsA*, which was subsequently transformed into *E. coli* BL21(DE3).

### 2.3. Expression, Purification, and Zymogram Analysis of the MIsA

*E. coli* BL21(DE3) cells harboring pET-29a-*MIsA* were cultivated in LB medium supplemented with 50 µg mL^–1^ kanamycin at 37 °C until an OD_600_ of 0.6 was reached. Protein overexpression was induced with 0.2 mM IPTG at 16 °C for 24 h. Harvested cells were lysed by ultrasonication in 50 mM Tris-HCl (pH 7.0) and centrifuged at 12,000× *g* for 30 min at 4 °C. The resulting supernatant was subjected to ammonium sulfate precipitation and hydrophobic chromatography. The purified MIsA was analyzed by sodium dodecyl sulfate-polyacrylamide gel electrophoresis (SDS-PAGE) [17] and visualized using Coomassie blue staining [18].

Zymogram analysis was performed using Native PAGE with a 10% acrylamide gel containing 0.2% (*w*/*v*) amylopectin [19]. Following electrophoresis, the PAGE gel was divided into two parts. One part was stained with Coomassie brilliant blue staining solution. The other part was incubated in 50 mM PBS (pH 6.0) at 50 °C for 10 min, then immersed in a dilute iodine solution for 10 min to develop the color.

### 2.4. Enzyme Activity Assay

The activity of the starch debranching enzyme was determined by quantifying the reducing sugars produced during starch hydrolysis [15]. The enzyme reaction mixture, consisting of an appropriate amount of enzyme and 0.5% (*w*/*v*) amylopectin in PBS buffer (50 mM, pH 6.0), was incubated at 50 °C for 10 min. The reducing sugars released were quantified using the dinitro salicylic acid (DNS) method [20]. One unit of enzyme activity was defined as the amount releasing 1 μmol of reducing sugar per minute. The protein content was determined using the Bradford method [21].

### 2.5. Biochemical Characterization of MIsA

The pH dependence of MIsA activity was investigated using various buffers: citrate buffer (50 mM, pH 3.0–6.0), sodium phosphate buffer (50 mM, pH 6.0–8.0), and Tris-HCl buffer (50 mM, pH 8.0–9.0), with starch as the substrate at a constant temperature of 50 °C. To assess pH stability, enzyme aliquots were incubated at pH values ranging from 3.0 to 10.0 for 12 h at 4 °C, followed by measurement of enzyme activity under standard conditions. The effect of temperature on enzyme activity was examined by incubating the enzyme with starch for 10 min at temperatures between 20 °C and 50 °C. Thermal stability was evaluated by incubating enzyme preparations at various temperatures. Samples were collected at specified time intervals, and residual activity was measured using the standard enzyme assay conditions. Unheated enzyme served as a control (100% activity).

### 2.6. Effect of Metal Ions and Chemicals on the Enzyme Activity of MIsA

The influence of metal ions and chemical compounds on MIsA activity was investigated by incubating the enzyme with various metal salts at 50 °C for 10 min. Additionally, the impact of chemical agents and solvents was assessed by exposing the enzyme to 10 mM EDTA, 5% (*v*/*v*) dimethyl sulfoxide (DMSO), isopropanol, methanol, ethanol, acetone, 10% (*v*/*v*) acetonitrile, and different concentrations of surfactants (20 mg mL^–1^ Triton X-100, DTT (1 mM, 10 mM), and 2 mg mL^–1^ SDS) under identical conditions. Enzyme activity in the absence of additives was defined as 100%.

### 2.7. Substrate Specificity of MIsA

The substrate specificity of MIsA was examined using various carbohydrates as alternative substrates. Enzyme assays were conducted in reaction mixtures containing 0.5% (*w*/*v*) of the following polysaccharides: α-cyclodextrin, tapioca starch, potato starch, soluble starch, maize starch, amylose, amylopectin, and pullulan in 50 mM phosphate-buffered saline (PBS) at pH 6.0.

### 2.8. Kinetic Parameters of MIsA

The purified enzyme was assayed with varying concentrations of amylopectin (0.1–1.4% *w*/*v*) and amylose (0.1–1.1% *w*/*v*) under optimized conditions (50 °C, pH 6.0, 10 min). Kinetic parameters, *V*_max_ and *K*_m_, were determined by fitting the data to the Michaelis–Menten equation [22].

### 2.9. Scanning Electron Micrographs of MIsA-Treated Potato Starch

Scanning electron microscopy (SEM) was utilized to examine the structural changes in MIsA-treated potato starch. A 10% (*w*/*v*) potato starch suspension was incubated with MIsA for various durations (0 h, 4 h, 9 h), followed by centrifugation at 12,000 rpm for 20 min. The resulting precipitate was lyophilized, pulverized, and dried at 45 °C for 48 h. For SEM analysis, the lyophilized powder was mounted on the sample stage using conductive double-sided adhesive tape and coated with a 30 nm gold layer under vacuum before examination.

### 2.10. Distribution of the Degree of Polymerization of Products Hydrolyzed by MIsA

The purified starch debranching enzyme was added to a solution of amylopectin at a final concentration of 0.5% (*w*/*v*). Amylopectin was prepared with Na_2_HPO_4_-NaH_2_PO_4_ buffer at pH 6.0. Reactions were conducted at 50 °C for 0.5 h and 1 h. Following the reaction, the enzyme was inactivated, and the mixture was centrifuged before lyophilizing the supernatant.

High-performance anion-exchange chromatography (HPAEC) analysis was performed on the sample extracts using a CarboPac PA-200 anion-exchange column (4.0 × 250 mm; Dionex) coupled with a pulsed amperometric detector (PAD; Dionex ICS 5000 system), with technical support provided by Sanshu Biotech. Co., Ltd (Suzhou, China). Chromatographic conditions included a flow rate of 0.4 mL min^–1^ and an injection volume of 5 μL. The solvent system consisted of 0.2 M NaOH and 0.2 M sodium acetate (NaAc) buffer. The gradient program was as follows: 0 min, 90:10 (NaOH:NaAc); 10 min, 90:10 (NaOH:NaAc); 30 min, 40:60 (NaOH:NaAc); 50 min, 40:60 (NaOH:NaAc); and returning to 90:10 (NaOH:NaAc) until 60 min [23,24].

### 2.11. Determination of the Glycosidic Bond Ratio

Of the MIsA, 5 mL was mixed with 10 mL of a 5% (*w*/*v*) amylopectin solution in 20 mM Tris-HCl (pH 7.0). The mixture was incubated at 50 °C for 10, 30, or 60 min. The reaction was terminated by boiling for 5 min, and the samples were then freeze-dried. An appropriate amount of purified starch (about 5 mg) was weighed into an EP tube, and 1 mL of DMSO was added. The mixture was heated at 80 °C overnight. After cooling, the supernatant was collected by centrifugation at 12,000 rpm for 10 min and transferred to an NMR tube for analysis [25].

### 2.12. Analysis of the MIsA Hydrolysis Pattern

The purified enzyme was added to solutions containing 1% (*w*/*v*) maltotetrasaccharide, maltopentasaccharide, maltohexaosaccharide, amylopectin, pullulan polysaccharide, amylose, and β-cyclodextrin, prepared in 20 mM Na_2_HPO_4_-NaH_2_PO_4_ buffer (pH 6.0). The reactions were conducted at 50 °C for 9 h. Post-reaction, mixtures were centrifuged (12,000 rpm, 2 min), and supernatants were filtered through a 0.22 μm membrane.

Thin-layer chromatography (TLC) analysis was performed using glucose and malt oligosaccharide standards (G2–G6). The TLC plate was developed with a solvent system consisting of n-butanol/methanol/water (4:2:1) and subsequently stained with a color developer comprising concentrated sulfuric acid/methanol (1:9) [26].

### 2.13. Determination of the Debranching Degree

MIsA was added to a 1% (*w*/*v*) amylopectin solution, which was incubated at 50 °C for various time intervals (0 h, 0.25 h, 0.5 h, 1 h, 3 h, and 9 h) [27]. The degree of starch debranching was determined via the equation (R_sample_ − R_amylopectin_)/(R_debranched_ − R_amylopectin_)] × 100, where R_sample_ represents reducing sugars in a sample debranched for a specific time, R_debranched_ represents reducing sugars in potato starch debranched for 9 h, and R_amylopectin_ represents reducing sugars in raw waxy starch [15].

### 2.14. Dual-Enzyme Coupling to Improve the Oligosaccharide Yield

A previously cloned α-amylase, designated MTHase, from *Myxococcus* sp. V11, which primarily hydrolyzes amylose to produce maltotetraose, was investigated. To examine the effect of combining MIsA with MTHase on maltotetraose yield, a substrate mixture containing 0.25% (*w*/*v*) amylose, amylopectin, and soluble starch in 20 mM Na_2_HPO_4_-NaH_2_PO_4_ buffer at pH 6.5 was incubated with the enzyme complex (MIsA:MTHase = 2:1) at 45 °C for 9 h, maintaining pH 6.5 [28]. The oligosaccharide yield was analyzed using high-performance liquid chromatography (HPLC) with a Cosmosil Sugar-D column (Nacalai Tesuque, Kyoto, Japan) and a refractive index detector (RID) at 30 °C. The mobile phase consisted of acetonitrile and water (65:35, v/v) at 1 mL min^–1^ flow rate. Glucose, maltose, maltotriose, and maltotetraose standards were used for quantification of released products.

## 3. Results

### 3.1. Cloning of the Debranching Enzyme Gene MIsA from Myxococcus sp. Strain V11

The cloned *misa* gene was 2139 bp in length, encoding a debranching enzyme comprising 712 amino acids. The calculated molecular weight of the MIsA protein was 80.45 kDa, with a predicted isoelectric point (pI) of 6.04.

Phylogenetic analysis (Figure 1) revealed that MIsA clusters with glycogen operon protein GlgX (NCBI accession number P0A4Y5.1) from *Mycobacterium tuberculosis* AF2122/97. Sequence alignment showed 54.7% identity with glycogen operon protein GlgX [29]. Notable similarities were also observed with TreX (48.94%, 2VNC A) from *Saccharolobus solfataricus* [30] and glycogen debranching enzyme from *Erwinia tasmaniensis* Et1/99 (47.26%, B2VJR7.1) [31]. MIsA shared less than 47% sequence identity with other bacterial debranching enzymes in GenBank. The phylogenetic tree indicates that starch debranching enzymes from *Myxococcus* sp. strain V11 form a distinct clade, suggesting MIsA represents a novel starch debranching enzyme.

The structural model of MIsA was generated using AlphaFold 3 (Figure 2a). MIsA contains four conserved sequence segments: I (^275^DVVYNH^280^), II (^348^GFRFDLA^354^), III (^388^EPWD^391^), and IV (^455^FVTAHD^460^), characteristic of the α-amylase family [32]. The MIsA structure comprises three domains: N, A, and C. Domain A is the catalytic (*β/α*)_8_-barrel domain, harboring three conserved catalytic residues (D352, E388, D460). Domain C exhibits a *β*-sandwich structure, while domain N is classified as a carbohydrate-binding module family 48 (CBM48) based on primary sequence alignment, indicating membership in the GH13_11 subfamily [2].

The MIsA monomer structure closely resembles that of TreX (RMSD = 0.779), a bifunctional glycogen-debranching enzyme from *Sulfolobus solfataricus*. Catalytic residues and substrate-binding amino acid residues, such as R350, W390, and Y397, show similarity to those in TreX. Four amino acid residues—H215, I216, F221, and D242—associated with metal ion binding are situated within Domain A. Notably, MIsA features a distinct helix 4 structure (amino acids 217–227) extending from the substrate-binding groove (Figure 2b), which enhances stability and activity on branched substrates with longer maltooligosaccharides [30]. In contrast to TreX, where Glu^94^ and Asp^318^ form a structural lid associated with *α*-1,4-transferase activity, MIsA contains Lys^86^ and Lys^307^ at corresponding positions [33].

### 3.2. Expression, Purification, and Zymogram Analysis of MIsA

MIsA from *Myxcococcus* sp. strain V11 was successfully expressed in *E. coli* BL21(DE3) and subsequently purified via Ni^+^ affinity chromatography. The MIsA confirmed an apparent molecular weight of about 80 kDa (Figure 3a). Native-PAGE analysis of the crude enzyme extract revealed two bands at the same position as those observed via SDS-PAGE (Figure 3b), indicating the presence of the same protein in both preparations.

### 3.3. Enzymatic Characterization

The activity of MIsA was observed in the neutral to moderately thermal temperature range (20–60 °C) (Figure 4a), with optimal activity at 50 °C (Figure 4a). Notably, MIsA exhibited a thermostimulatory effect, where activity increased after incubation at 20–50 °C for 3 h. Significant thermostimulation was observed after 3 h at 30 °C and 50 °C, with elevated relative activity after 1 h incubation across various temperatures compared to the control. However, MIsA activity decreased after 9 h incubation at 50 °C, with 63.95% residual activity remaining after 12 h incubation at 40 °C (Figure 4b). MIsA also demonstrated activity across a broad pH range (5.0–8.0), with optimal activity at pH 6.0 (Figure 4c). The enzyme retained over 60% of its original activity across a wide pH range of 4.0 to 10.0 (Figure 4d).

### 3.4. Effects of Metal Ions and Chemical Reagents on MIsA Activity

The effects of various metal ions on MIsA activity were investigated (Table 1). Most metal ions exhibited a significant inhibitory effect. Notably, 5 mM Ca^2+^ completely inhibited MIsA activity, indicating that its activity is calcium independent. Cu^2+^, Co^2+^, Fe^2+^, Zn^2+^, Ni^2+^, and Fe^3+^ significantly inhibited MIsA at concentrations of 1 mM and 5 mM, while Ni^2+^ and Cr^3+^ showed inhibition only at 5 mM. In contrast, Na^+^ had no effect at 1 mM and 5 mM concentrations. Mg^2+^ showed no significant effect at 1 mM but increased MIsA activity by 50% at 5 mM. Additionally, Li^+^ emerged as an effective activator, enhancing MIsA activity by over 20%.

MIsA was strongly inhibited by 5 mg mL^–1^ SDS and 10% acetonitrile. However, the enzyme remained highly stable in the presence of nonionic surfactants, retaining full activity with 10% methanol (Table 2). Residual enzyme activities were 75.79%, 58.84%, 55.89%, and 33.07% following treatment with EDTA, ethanol, isopropanol, and acetone, respectively. Low concentrations of DTT (1 mM, 10 mM), Triton X-100 (20 mg mL^–1^), and DMSO (5%) did not affect MIsA activity.

### 3.5. The Activity of MIsA to Different Substrates 

The substrate specificity of the recombinant MIsA was evaluated under standard conditions using various carbohydrates at a final concentration of 0.5 mg mL^–1^. The substrates tested included amylopectin, amylose, pullulan, soluble starch, potato starch, tapioca, corn starch, α-cyclodextrin, and β-cyclodextrin (Table 3). Among these substrates, amylopectin was found to be the most suitable substrate for MIsA. Significant hydrolysis was observed for amylose, while partial hydrolysis occurred with corn starch. In contrast, no hydrolysis was detected for maize starch, potato starch, tapioca starch, pullulan, soluble starch, α-cyclodextrin, and β-cyclodextrin. The yields of amylopectin and amylose hydrolyzed into oligosaccharides were 10.16% and 11.70% (Appendix A).

### 3.6. Kinetic Analysis of MIsA

The kinetic parameters of MIsA were measured at pH 6.0 and 50 °C with amylopectin and amylose as substrates. According to the Michaelis–Menten equation (Figure 5), the *K*_M_, *K*_cat_, and *V*_max_ values of MIsA for amylopectin were 1.22 mM, 40.42 µmol min^–1^ mg^–1^, and 4064.31 mM·min^–1^, respectively (Figure 5a), and the *K*_M_, *K*_cat_, and *V*_max_ values of MIsA for amylose were 2.53 mM, 14.77 µmol·min^–1^ mg^–1^, and 2902.97 mM·min^–1^, respectively (Figure 5b).

### 3.7. Morphological Observation of Starch Granules After MIsA Treatment

Compared with that of untreated potato starch, the microstructure of potato starch treated with MIsA was notably different. The natural potato starch granules were smooth (Figure 6a). In contrast, after treatment with MIsA for 4 h (Figure 6b) or 9 h (Figure 6c), fissures became evident on the surfaces of the starch granules. Additionally, flocculation was observed, and with increasing treatment time, both the number and size of fissures on the granule surfaces increased progressively.

### 3.8. Polymerization Degree Distribution of Starch Treated with MIsA

To examine the chain length distribution within starch, a chain length distribution curve was generated by plotting the degree of polymerization (DP) value on the *x*-axis and the relative peak area corresponding to each DP value on the *y*-axis [34]. The debranching efficacy of MIsA was assessed and compared to Promozyme D2, an isoamylase from *Pseudomonas* spp. [14]. Samples treated with Promozyme D2 for 24 h (line a) and MIsA for 30 min (line b) and 1 h (line c) exhibited increased chains in the DP 6–13 range and decreased chains in the DP 14–50 range. Notably, the polymerization degree distribution primarily fell within the DP 5–30 range, with branched chains exceeding DP 33 comprising less than one-tenth of the total. Analysis of MIsA-treated samples revealed that shorter reaction times yielded shorter chains, while prolonged incubation periods resulted in MIsA primarily acting on shorter chains, producing less polymerized products (Figure 7 and Appendix A).

### 3.9. Determination of Glycosidic Bond Ratios

The glycosidic bond distribution was analyzed using 1H NMR spectroscopy (Figure 8). The chemical shifts of the anomeric protons corresponding to α-1,4 and α-1,6 glycosidic bonds were 5.12 ppm and 4.78 ppm, respectively, falling within the expected range. Quantification revealed that α-1,6 glycosidic bonds accounted for 7.24% of the total glycosidic bonds in the control group. Following MIsA treatment, the H-1(1–6) proton peak intensity decreased, yielding 6.16%, 5.60%, and 5.47% α-1,6 glycosidic bonds after 10 min, 30 min, and 60 min, respectively. These results demonstrate that MIsA reduces the branching degree of branched amylose by specifically hydrolyzing α-1,6 glycosidic bonds, confirming its isoamylase activity.

### 3.10. Catalytic Properties of the MIsA

The mode of action of MIsA on various substrates, including linear malt oligosaccharides (G4–G6), amylopectin, amylose, pullulan polysaccharides, and β-cyclodextrin, was examined using thin-layer chromatography (TLC) analysis (Figure 9). The results showed that the starch debranching enzyme MIsA hydrolyzed pullulan to maltotriose and failed to hydrolyze β-cyclodextrin. In contrast, hydrolysis of amylopectin, amylose, and linear malt oligosaccharides (G4–G6) yielded linear oligosaccharides, along with glucose and malt oligosaccharides (G2–G6). These findings suggest that MIsA facilitates the intermolecular transfer of glucans through the formation of an α-1,4-glucosidic linkage between 1,4-α-D-glucan molecules, as well as hydrolysis of the α-1,6-glycosidic linkage. Consequently, MIsA exhibited disproportionation activity toward maltooligosaccharides, characteristic of a 4-α-GTase (EC 2.4.1.25) [35,36,37].

### 3.11. Degree of Debranching of MIsA

The degree of debranching reached 61.3% after 0.5 h and 94.14% after 3 h, and the maximum debranching percentage reached 100% at 9 h (Table 4). Compared with IsoM, MIsA demonstrated greater debranching efficiency, requiring less time to achieve the same level of debranching (Appendix A).

### 3.12. Dual Enzyme Coupling to Improve the Reducing Sugar Yield

Hydrolysis of amylose, amylopectin, and soluble starch by the composite enzyme mixture yielded 67.8%, 76.81%, and 70.01% reducing sugars, respectively. In comparison, hydrolysis by MIsA alone produced 28.5%, 56.22%, and 42.92% reducing sugars, whereas MTHase alone yielded 34.6%, 46.01%, and 37.8% reducing sugars, respectively (Figure 10). Compared with that of MIsA alone, the activity of the composite enzyme mixture was 2.38-, 1.37-, and 1.63-fold greater, and that of MTHase alone was 1.96-, 1.67-, and 1.85-fold greater, respectively. Additionally, the composite enzyme mixture (MIsA + MTHase) produced 2.7, 1.41, and 1.97 times more maltotetrasaccharide than MTHase did and 1.8, 1.29, and 1.33 times more maltotetrasaccharide than MIsA did.

## 4. Discussion

The starch debranching enzymes have garnered considerable attention from both academic and industrial researchers due to their versatile applications in starch processing. This study focused on characterizing MIsA, a bifunctional starch debranching enzyme derived from *Myxococcus* sp. strain V11. Heterologous expression and purification in *E. coli* yielded a specific activity of 317.31 U·mg^–1^, without requiring Li^+^ supplementation. MIsA’s optimal performance was observed at 50 °C, with activity maintained across a broad pH range, from acidic to alkaline, similar to amylopullulanase (ApuNP1) from *Geobacillus thermoleovorans* NP1 and iam from *Flavobacterium* sp. [38,39] (Table 5). Unlike IsoM from *Corallococcus* sp. strain EGB, which is activated by 1 mM Ca^2+^ [15], MIsA’s activity is Ca^2+^-independent. However, Li^+^ and Mg^2+^ serve as effective activators, enhancing MIsA’s activity by over 20% and about 56%, respectively, at 5 mM. Notably, MIsA exhibits a thermostimulatory effect, where activity increases following incubation at 20–50 °C for 3 h. This unique property distinguishes MIsA from other reported bifunctional starch debranching enzymes.

Chain length distribution analysis of amylopectin revealed that MIsA preferentially hydrolyzes short-chain glucans, which aligns with the mode of action of the isoamylase SU1 from *Zea mays* [45]. The debranching pattern of MIsA, as determined by 1H-NMR spectroscopy, indicates that MIsA specifically hydrolyzes α-1,6-glycosidic bonds within the polymer, which is consistent with the action of the pullulanase ZPU1 [46]. Micromorphological observations of starch granules and HPAEC analysis further confirmed the effective debranching activity of the enzyme.

MIsA effectively hydrolyzed pullulan, a linear polysaccharide with α-1,6-linked maltotriose units, into maltotriose. Additionally, it generates a series of maltooligosaccharides and glucose from shorter maltooligosaccharides. MIsA also produces linear oligosaccharides from amylose and amylopectin.

Type I amylopullulanase specifically cleaves α-1,4-glycosidic bonds, whereas Type II amylopullulanase hydrolyzes both α-1,4 and α-1,6-glycosidic bonds. Additionally, 4-α-glucanotransferase (4-α-GTase) catalyzes the disproportionation process, involving the transfer of a glucan chain from one glucan molecule’s non-reducing end to another’s [39,47]. Notably, MIsA exhibits both Type II amylopullulanase and 4-α-glucanotransferase activities. Site mutation results indicate that these three catalytic activities share the same active center (Appendix A). This dual functionality is unprecedented, as previously reported enzymes lack this combination. For example, isoamylase IsoM from *Corallococcus* sp. strain EGB cannot hydrolyze pullulan and displays limited glycogen hydrolysis, with only 22% activity relative to its optimal substrate [15]. Furthermore, the amylopullulanase (gt-apu) from *Geobacillus thermoleovorans* NP33 possesses a single active site for α-amylase and pullulanase activities, as demonstrated by enzyme kinetic analyses using competing substrates and inhibitors [48].

The hydrolysis of maltooligosaccharides by MIsA, yielding a range of maltooligosaccharides and glucose, aligns with observations reported for Trex from *Sulfolobus solfataricus* ATCC 35092 [45]. Similar to Trex, MIsA exhibits both isoamylase and 4-alpha-glucanotransferase activities. However, unlike Trex, MIsA demonstrates the ability to hydrolyze both amylose and amylopectin, indicating dual endo- and exo-acting enzyme activities. This distinction highlights MIsA’s versatility in starch degradation. This result is consistent with the production of the amylopullulanase GlgU from *Lactobacillus crispatus*. GlgU is a type II amylopullulanase that has only one catalytic structural domain with substrate specificity for α-1,4 and α-1,6-glycosidic bonds and is able to completely hydrolyze glycogen into maltooligosaccharides [46]. To the best of our knowledge, no other multifunctional starch debranching enzymes exhibit the diverse characteristics observed in MIsA. Consequently, further investigations into the multifunctionality of MIsA are of considerable scientific interest.

Owing to its multifunctionality, MIsA combines with the amylase MTHase to increase the yield of reducing sugars. In this study, both MIsA and α-amylase MTHase were used to increase the conversion of starch to maltotetrasaccharide to increase the productivity of maltotetrasaccharide. The results showed that the composite enzyme mixture (MIsA + MTHase) produced 2.7 times more maltotetrasaccharide than did MTHase and 1.8 times more maltotetrasaccharide than did MIsA. The presence of α-1,6-glycosidic bonds hinders the effective hydrolysis of starch by enzymes. When the synergistic action of the debranching enzyme and amylase is used at the same substrate concentration, the glucose value can reach 97.0%, the raw material utilization rate can increase by 0.3–0.5%, the reaction time can be shortened by 15%, and the dosage of the amylase enzyme can be reduced by 15%, which can save more than 30,000 tons of starch raw material per year [9]. The synergistic effect with amylase can shorten the process time, reduce the amount of enzyme added, reduce the demand for raw materials, and have high industrial application value.

In summary, MIsA is a novel multifunctional starch debranching enzyme with distinctive endo-, exo-, and transglycosylation activities, distinguishing it from other debranching enzymes within the GH13 family. MIsA’s capabilities enhance starch hydrolysis and prevent limiting dextrin formation during branched starch processing, thereby significantly improving starch raw material utilization and conversion rates. Its unique functional profile makes MIsA a valuable asset in the starch industry, holding considerable promise for advancing starch processing technologies and applications.

## Figures and Tables

**Figure 1 foods-13-03481-f001:**
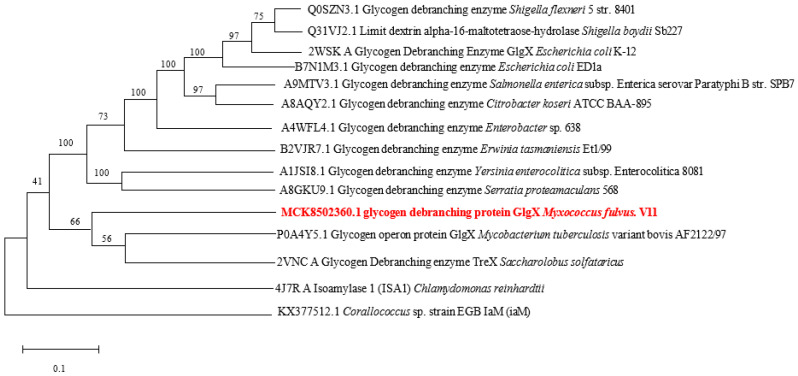
Multiple sequence alignment of the starch debranching enzyme *MIsA* from *Myxococcus* sp. strain V11 with other starch debranching enzymes from various organisms. (Bold red text represents the starch debranching enzyme *MIsA* from *Myxococcus* sp. strain V11).

**Figure 2 foods-13-03481-f002:**
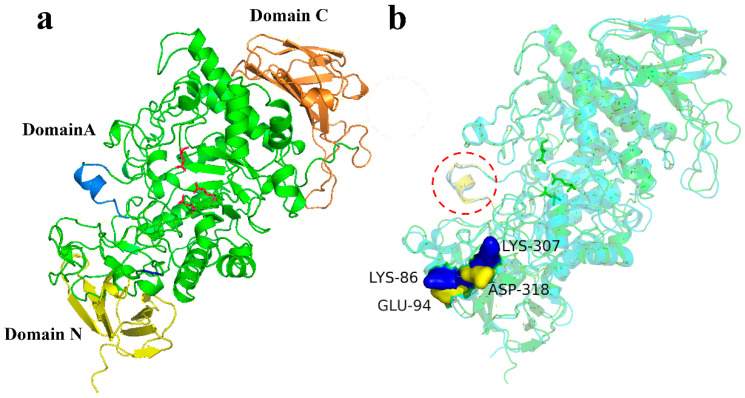
Comparison of the structures of TreX and MIsA. (**a**). Overall structure of MIsA; red sticks indicate catalytic amino acid residues. (**b**). Comparison of the structures of TreX (cyan) and MIsA (green). The red circle represents helix 4, the lids of TreX (yellow) and MIsA (blue).

**Figure 3 foods-13-03481-f003:**
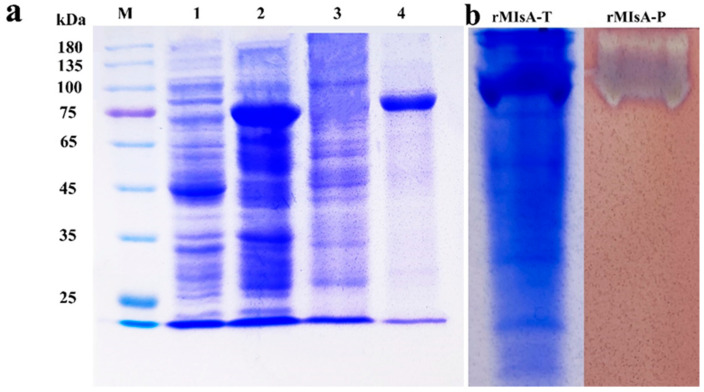
SDS-PAGE gel of MIsA. (**a**) Analysis of the expression of MIsA via SDS-PAGE. Lane M: standard protein marker; Lane 1: pET-29a (+) empty crude enzyme mixture; Lane 2: MIsA expression strain crude enzyme mixture; Lane 3: MIsA purified by Ni^2+^-affinity chromatography; Lane 4: MIsA purified by dialysis. (**b**) MIsA was analyzed by Native-PAGE. Lane 1: MIsA expression strain crude enzyme supernatant; lane 2: zymogram analysis of MIsA visualized by activity staining.

**Figure 4 foods-13-03481-f004:**
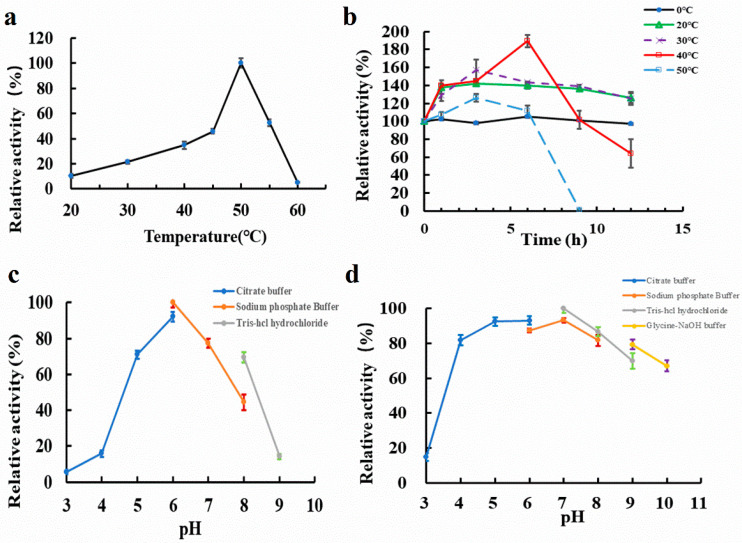
Effects of temperature and pH on the activity and stability of MIsA. (**a**) Optimal temperature. Activity was measured in 50 mM PBS buffer (pH 6.0) for 10 min. (**b**) Thermal stability. The activity was measured in 50 mM PBS buffer (pH 6.0) for 10 min after incubation of the enzyme at different temperatures for 12 h. (**c**) The optimal pH. Assays were carried out at 50 °C for 10 min in buffers of varying pH values (3.0–9.0). (**d**) pH stability. The activity was measured in 50 mM PBS (pH 6.0) at 50 °C for 10 min after the purified enzyme was incubated with buffers of various pH values at 4 °C for 12 h.

**Figure 5 foods-13-03481-f005:**
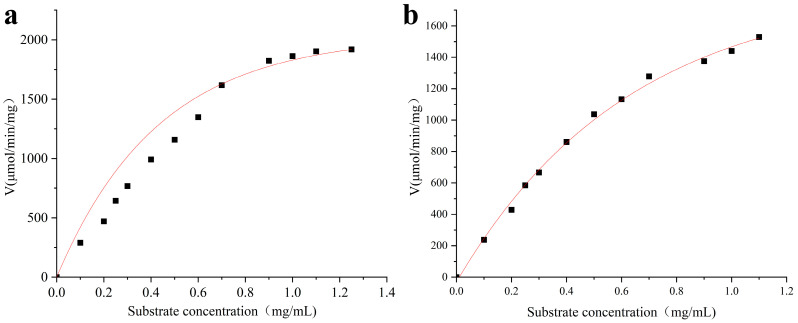
Enzyme kinetics of MIsA. (**a**) Amylopectin as the substrate. (**b**) Amylose as the substrate.

**Figure 6 foods-13-03481-f006:**
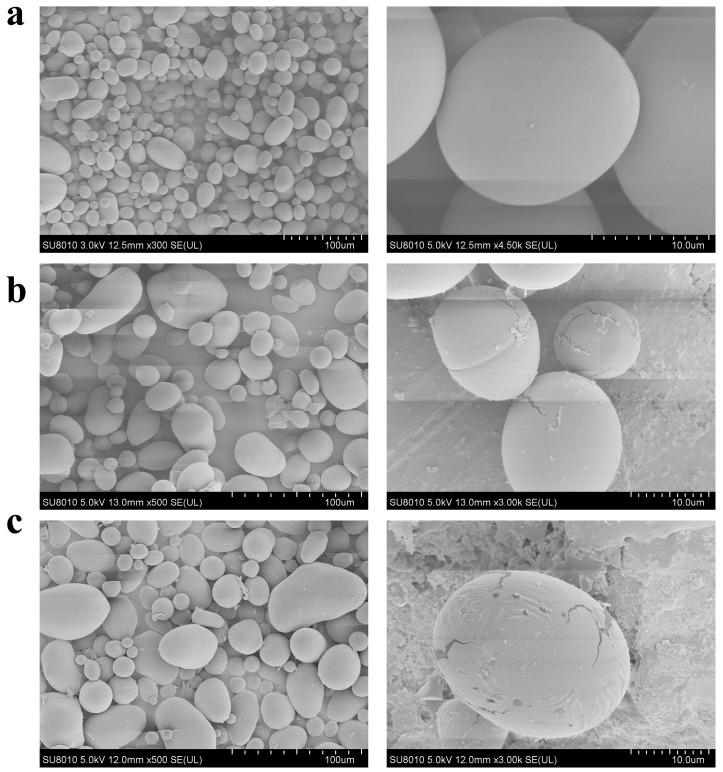
SEM analysis of insoluble starch granules. (**a**) Nondebranched starch; (**b**) starch debranched for 4 h; (**c**) starch debranched for 9 h. Pictures are shown at low (1 mm) and high (0.01 mm) magnification.

**Figure 7 foods-13-03481-f007:**
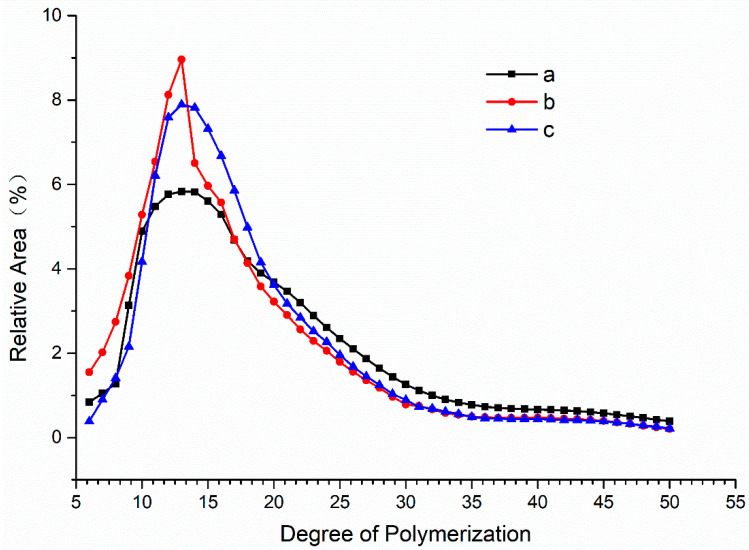
Chain length distribution (CLD) of amylopectin. (**a**) CLD of amylopectin after treatment with Promozyme^®^ D2 for 24 h. (**b**) CLD of amylopectin after treatment with MIsA for 0.5 h. (**c**) CLD of amylopectin after treatment with MIsA for 1 h.

**Figure 8 foods-13-03481-f008:**
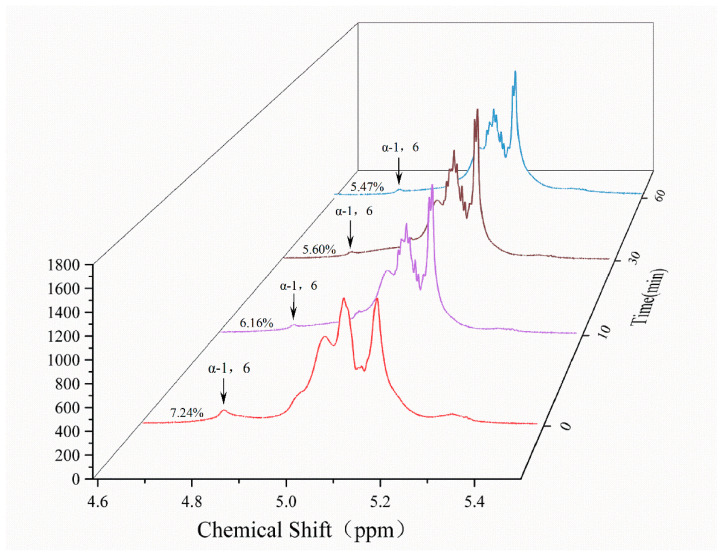
Determination of glycosidic bond ratios. The number of α-1,6 glycosidic bonds in the control group accounted for 7.24% of the total glycosidic bonds. After 10 min, 30 min, and 60 min of treatment with MIsA, the number of α-1,6 glycosidic bonds are 6.16%, 5.60%, and 5.47%, respectively.

**Figure 9 foods-13-03481-f009:**
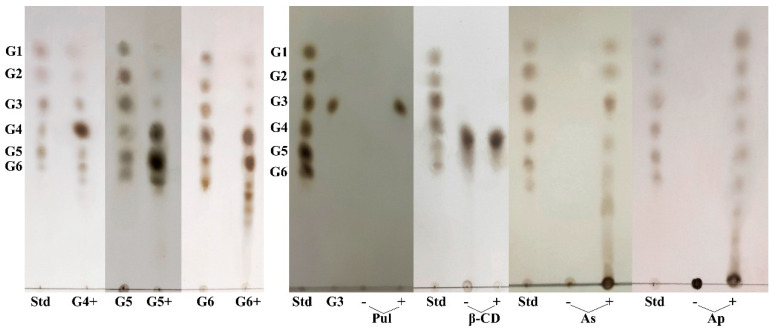
TLC analysis of MIsA activity toward various substrates. MIsA was reacted in 50 mM sodium phosphate buffer (pH 6.0) at 50 °C for 9 h with each substrate. G4, maltotetraose; G5, maltopentaose; G6, maltohexarose; β–CD, β–cyclodextrin; Pul, pullulan; AP, amylopectin; As, amylose; Std, standard of glucose and maltooligosaccharides (G2–G6); reactions on substrates with (+) and without (–) enzymes. The developing solvent was n-butanol/methanol/water at a ratio of 4:2:1 (*v/v/v*).

**Figure 10 foods-13-03481-f010:**
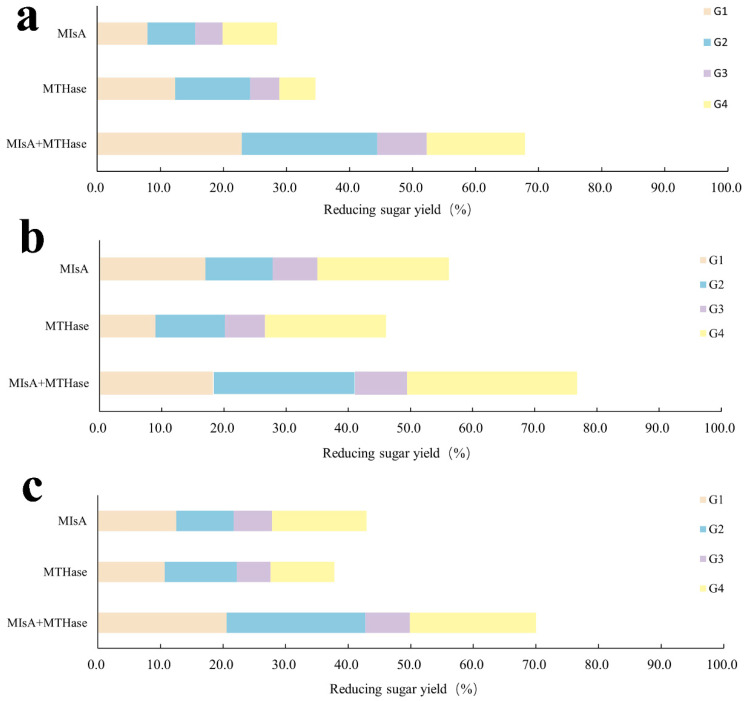
The oligosaccharide yields of different substrates treated with MIsA and MTHase. (**a**) Amylose; (**b**) amylopectin; (**c**) soluble starch.

**Table 1 foods-13-03481-t001:** Effects of metal ions on the enzyme activity of MIsA ^α^.

Metal Ion (1 mM)	Relative Activity (%) ± SD	Metal Ion (5 mM)	Relative Activity (%)
Li^+^	119.78 ± 2.70 ^a^	Mg^2+^	156.89 ± 2.00 ^a^
No addition	100 ± 1.09 ^b^	Li^+^	138.52 ± 1.48 ^b^
Cr^3+^	98.09 ± 0.69 ^b^	K^+^	116.89 ± 2.25 ^c^
Na^+^	94.59 ± 1.02 ^c^	No addition	100 ± 1.09 ^d^
Mg^2+^	92.09 ± 1.58 ^d^	Na^+^	93.89 ± 0.67 ^e^
K^+^	90.18 ± 0.34 ^d^	Ni^2+^	0 ± 3.79 ^f^
Ni^2+^	63.54 ± 2.62 ^e^	Cu^2+^	0 ± 1.07 ^f^
Ca^2+^	54.78 ± 3.57 ^f^	Fe^2+^	0 ± 4.98 ^f^
Fe^2+^	0 ± 1.23 ^f^	Co^2+^	0 ± 3.00 ^f^
Co^2+^	0 ± 0.20 ^f^	Fe^3+^	0 ± 1.56 ^f^
Fe^3+^	0 ± 1.00 ^f^	Ca^2+^	0 ± 1.23 ^f^
Cu^2+^	0 ± 2.01 ^f^	Zn^2+^	0 ± 5.12 ^f^
Zn^2+^	0 ± 6.07 ^f^	Cr^3+^	0 ± 2.96 ^f^
Mn^2+^	0 ± 0.74 ^f^	Mn^2+^	0 ± 2.94 ^f^

^α^ Mean values ± standard deviations from three independent experiments are shown. The letters indicate a significant difference of 0.05.

**Table 2 foods-13-03481-t002:** Effects of chemicals on the enzyme activity of MIsA ^α^.

Chemical Reagents	Concentration	Relative Activity (%) ± SD
DTT	1 mM	102.5 ± 3.69 ^a^
No addition	0	100 ± 3.64 ^a^
Methanol	10%	99.27 ± 3.13 ^a^
Triton X-100	20 mg mL^–1^	96.89 ± 3.45 ^a^
DTT	10 mM	94.8 ± 4.05 ^a^
Triton X-100	20 mg mL^–1^	92.89 ± 4.10 ^a^
EDTA	10 mM	75.79 ± 1.20 ^b^
Ethanol	10%	58.84 ± 4.75 ^c^
Isopropanol	10%	55.89 ± 0.45 ^c^
Acetone	10%	33.07 ± 4.56 ^d^

^α^ Mean values ± standard deviations from three independent experiments are shown. The letters indicate a significant difference of 0.05.

**Table 3 foods-13-03481-t003:** Substrate specificity of MIsA ^α^.

Substrate	Specific Activity (U/mg) ± SD	*K*_m_ (mM)
Amylopectin	317.31 ± 2.34 ^a^	1.22
Soluble Starch	209.71 ± 2.89 ^b^	-
Potato Starch	197.59 ± 4.06 ^c^	-
Tapioca	150.76 ± 3.83 ^d^	-
Corn Starch	148.56 ± 4.44 ^d^	-
Pullulan	76.41 ± 4.57 ^e^	
Amylose	55.27 ± 4.28 ^f^	2.53
Alpha-Cyclodextrin	2.2 ± 1.56 ^g^	-
β-Cyclodextrin	0	-

^α^ Mean values ± standard deviations from three independent experiments are shown. The letters indicate a significant difference of 0.05.

**Table 4 foods-13-03481-t004:** The debranching degree of MIsA ^α^.

Time/h	Debranching Rate (%) ± SD
0	0
0.25	34.09 ± 1.94 ^e^
0.5	61.30 ± 1.15 ^d^
1	80.28 ± 1.45 ^c^
3	94.14 ± 0.23 ^b^
9	100.00 ± 0.15 ^a^

^α^ Mean values ± standard deviations from three independent experiments are shown. The letters indicate a significant difference of 0.05.

**Table 5 foods-13-03481-t005:** Comparison of the enzymatic properties of MIsA and other starch debranching enzymes.

										Relative Activity (%)						
Enzyme	Source	Molecular Size (kDa)	Optimal pH	Optimal Temperature (°C)	pH Stability	Thermal Stability (°C)	Amylopectin	Pullulan	Corn Starch	Potato Starch	Soluble Starch	Amylose	Glycogen	Optimal Substrate	K_m_	Reference
MIsA	*Myxococcus* sp. V11	80.45	6.0	50	4.0–10.0	0–55	100	-	47.8	62.3	66.1	17.4	-	Amylopectin	2.92 mM (DNS method)	This study
IAM	*Bacillus lentus* CICIM304	100	6.5	70	4.0–9.0	30–80	82	4	72	-	68	0	100	Ostreidae Glycogen	0.403 mg mL^−1^ (I_2_—KI solution)	[40]
IsoM	*Corallococcus* sp. EGB	90	6.0	45	3.0–11.0	20–70	20.9	0.6	87.8	100	19.5	0	22.1	Potato Starch	7.4 mg mL^−1^ (I_2_—KI solution)	[15]
treX	*Thermophilic archaeon Sulfolobus solfataricus* ATCC 35092	76	5.0	75	4.0–10.0	40–95	100	0	-	-	-	12	0.96	Amylopectin	-	[41]
iam	*Flavobacterium* sp.	83	6.0–7.0	40	4.0–9.0	20–50	-	-	-	-	-	-	-	Ostreidae Glycogen	-	[42]
PY35	*Pectobacterium chrysanthemi* PY35	74	7.0	40	3.0–10.0	10–90	100	-	-	-	-	-	80	Amylopectin	-	[43]
gt-apu	*Geobacillus thermoleovorans* NP33	182	7.0	60	-	-	-	-	34.33	51.79 (potato)	39.71 (rice)	-	-	Soluble Starch	-	[44]
Amylomaltase	*Thermus aquaticus* ATCC 33923	57.2	4.0–10.0	75	5.5–6.0	0-85	-	-	-	-	-	-	-	-	-	[37]
ApuNP1	*Geobacillus thermoleovorans* NP1	112 and 107	6.0	50	3.0–12.0	30–50	-	-	-	-	-	-	-	-	-	[38]

- Not available.

## Data Availability

The original contributions presented in the study are included in the article/Appendix A, further inquiries can be directed to the corresponding authors.

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
