# Peer review of "Molecular Cloning, Characterization, and Application of a Novel Multifunctional Isoamylase (MIsA) from *Myxococcus* sp. Strain V11"

_foods, 2024, doi:10.3390/foods13213481_

Round 1

Reviewer 1 Report

Comments and Suggestions for Authors

The research paper by Feng et al. is a fascinating study with many novel features discovered in the tri-functional glycosidase. The study has been performed methodically and the enzyme has been characterized in detail. Overall the data supports the conclusions.

I have some suggestions for improving the manuscript further. A couple of key novel features should be probed further and there are few weaknesses.

1.      The bioinformatics analysis (structure-Function relationship) is very weak. Keeping in view that the enzyme has novel features, a Clustal Multiple Alignment of key catalytic regions should be given, critical catalytic residues highlighted, and important amino acid substitutions should be discussed in the context of the enzyme's multiple functions.  Similarly, the Alphafold3 3D structure of the enzyme must be generated, and the multiple activities of the enzyme discussed. Key amino acids involved in activity should be highlighted. The MIsA structure can be superimposed on another enzyme that shows a single activity to highlight structural differences. Also, AF3 provides metal ion incorporation therefore the activation by Mg+2 can be dissected.

2.      Figure 3B is very interesting as at 40 deg C, the enzyme activity is doubled due to the thermostimulatory effect. This aspect should be examined further in the context of PRODUCTIVITY analysis. MTHase, MIsA and MTHase+MIsA should be subjected to productivity analysis at 40 and 50 deg C using both amylopectin and soluble starch in the absence and presence of Mg+2 for at least 12-15 h like in Figure 10. Kindly consult the following reference (Int. J. Mol. Sci. 202223(13), 6908; https://doi.org/10.3390/ijms23136908). Enzyme reaction rates over time are continuously influenced by protein unfolding due to temperature, rapidly changing reaction compositions and conditions, such as decreasing substrates and increasing product concentrations as well as their effect on enzyme activation and inhibition.  

3.      In Table 2, a thiol reagent like DDT should have been used to see if Disulfide bonds are present between any cysteine residues (if present in the enzyme sequence).

4.      The English needs to be improved as well as the overall presentation. The legends to figures need to be improved and details should be provided. For example, the Fig 4 legend is incomplete.

Comments on the Quality of English Language

Presentation and English needs improvement.

Author Response

  1. Bioinformatics analysis (structure-Function relationship) is very weak. Keeping in view that the enzyme has novel features, a Clustal Multiple Alignment of key catalytic regions should be given, critical catalytic residues highlighted, and important amino acid substitutions should be discussed in the context of the enzyme's multiple functions.  Similarly, the Alphafold3 3D structure of the enzyme must be generated, and the multiple activities of the enzyme discussed. Key amino acids involved in activity should be highlighted. The MIsA structure can be superimposed on another enzyme that shows a single activity to highlight structural differences. Also, AF3 provides metal ion incorporation therefore the activation by Mg2+ can be dissected.

Response: We analyzed the structure of MIsA. The structural model of MIsA was obtained using Alpha fold 3 (Fig. 2a). MIsA contains conserved sequence segments I (275DVVYNH280), II (348GFRFDLA354), III (388EPWD391), and IV (455FVTAHD460), which are common across the family of α-amylases (https://doi.org/10.1016/s0167-4838(00)00302-2). The MIsA structure comprises three domains: N, A, and C. Domain A is the catalytic (β/α)8-barrel domain, where three conserved catalytic residues (D352, E388, D460) were identified. Domain C exhibits a typical β-sandwich structure, while domain N is classified as a carbohydrate-binding module family 48 (CBM48) based on primary sequence alignment. These findings indicate that MIsA belongs to the GH13_11 subfamily (https://doi.org/10.1007/s00018-016-2241-y).

The overall structure of the MIsA monomer is similar to that of TreX (RMSD = 0.779), a bifunctional glycogen-debranching enzyme from Sulfolobus solfataricus. In addition to the catalytic residues, the substrate-binding amino acid residues of MIsA, such as R350, W390, and Y397, also resemble those in TreX. Four amino acid residues—H215, I216, F221, and D242—associated with metal ion binding are situated within Domain A. MIsA features a helix 4 structure that extends from the bottom of the substrate-binding groove, comprising amino acids 217 to 227 (Fig. 2b). This helix 4, which is absent in other isoamylases and pullulanases, enhances stability and increases activity on branched substrates with longer maltooligosaccharides (https://doi.org/10.1074/jbc.M802560200). The α-1,4-tranformsase activity of TreX is thought to be associated with two amino acid residues, Glu94 and Asp318, which form a structural lid. This indicates that the α-1,4-transferase activity may be affected by the oligomeric arrangement. In MIsA, the residues corresponding to Glu94 and Asp318 are Lys86 and Lys307, respectively (https://doi.org/10.1093/jb/mvj129).

Fig. 2 Comparison of the overall structure of TreX and MIsA

  1. Overall structure of MIsA, Red sticks indicate catalytic amino acid residues.
  2. Comparison of the structure of TreX(cyan) and MIsA(green). The red circle represents helix 4, the lids of TreX(yellow) and MIsA(blue)

  1. Figure 3B is very interesting as at 40 deg C, the enzyme activity is doubled due to the thermostimulatory effect. This aspect should be examined further in the context of PRODUCTIVITY analysis. MTHase, MIsA and MTHase+MIsA should be subjected to productivity analysis at 40 and 50 deg C using both amylopectin and soluble starch in the absence and presence of Mg+2 for at least 12-15 h like in Figure 10. Kindly consult the following reference (Int. J. Mol. Sci. 2022, 23(13), 6908; https://doi.org/10.3390/ijms23136908). Enzyme reaction rates over time are continuously influenced by protein unfolding due to temperature, rapidly changing reaction compositions and conditions, such as decreasing substrates and increasing product concentrations as well as their effect on enzyme activation and inhibition.

Response: Thank you for your suggestion and we have taken it into consideration. We have investigated the effects of substrate concentrations, pH and temperature on dual enzyme coupling to improve oligosaccharides yield in a one-way experiment. In the temperature stability experiments, MTHase was completely inactivated after 5h at 40°C and MIsA was completely inactivated after 9h at 50°C. Therefore, the final reaction time in the experiments to determine the dual enzyme coupling to improve oligosaccharides yield was 9h.

a: The highest temperature at which the dual enzyme coupling to produce reducing sugars is 45°C.

b: The highest pH at which the dual enzyme coupling to produce reducing sugars is PBS 6.5 (50mM).

c: The highest substrate concentration at which the dual enzyme coupling to produce reducing sugars is 0.25 mg/mL.

d: The addition of Mg2+ in the amount of 5 mM inhibited the activity of MTHase and reduced the amount of reducing sugar yield, so the addition of 5 mM of Mg2+ did not result in an increase in the amount of reducing sugar in the system of dual enzyme coupling. (Red underlining is with the addition of 5 mM of magnesium ions, blue is without).

a: Temperature (35℃-55℃).

b: pH (The ionic strength of all buffers was 50 mM).

c: Different substrate concentrations (0.1-2.0 mg mL-1).

d: Add Mg2+ and not.

Note: The reaction times for groups a, b and c are 15 min and d is 9h.

The oligosaccharide yields different substrates are treated with MIsA and MTHase. a: Amylose; b: Amylopectin; c: Soluble starch.

  1. In Table 2, a thiol reagent like DDT should have been used to see if Disulfide bonds are present between any cysteine residues (if present in the enzyme sequence).

Response: Through the secondary structure of MIsA, we found that MIsA has no disulfide bond, and we have verified the effect of DTT on MIsA activity. The data has been added to the manuscript.

Chemical reagents

Concentration

Relative activity (%) ± SD

DTT

1 mM

102.5±3.69a

No addition

0

100±3.64a

DTT

10 mM

94.8±4.05a

  1. The English needs to be improved as well as the overall presentation. The legends to figures need to be improved and details should be provided. For example, the Fig 4 legend is incomplete.

Response: The English have been checked and corrected by professional company. The legends to figures have been improved and provided details.

Reviewer 2 Report

Comments and Suggestions for Authors

The manuscript entitled “Molecular Cloning, Characterization, and Application of a Novel Multifunctional Isoamylase (MIsA) from Myxococcus sp. strain V11” was carefully revised. In general, the manuscript has a good technical quality and clarity of presentation. Howeever, some points need to be addressed:

1. Title: “Myxococcus” should always be italiczed. The same for Escherichia coli (line 12). I recommend correct/revise all manuscript.

2. How is this system different to other reports to merit publication? The novelty of this study should be highlighted in Introduction section.

3. The end of the introduction should be a remark of the interest of the study.

4. Line 143: The authors should express units for amylopectin. I recommend revise all manuscript.

5. Lines 151/166: The authors should report the ionic strenght used in all buffer solutions.

6. Fig. 3 and Tables 1, 2, and 3: What is 100% relative activity. I recommend add values for activity values at 100% in each Figure legend or Table.

7. Fig. 4: Linearization promotes the incorrect interpretation of models due to oversimplification. The authors should perform kinetic studies by fitting non-linear Michaelis-Menten model to the experimental data to determine apparent kinetic constant values. I recommend this study that confirms the superiority of parameter estimation (Km and Vmax) by nonlinear method over classical linearized model - https://doi.org/10.12793/tcp.2018.26.1.39 and https://doi.org/10.1021/acs.jchemed.6b00629 . Therefore, the authors should calculate such apparent kinetic parameters using a nonlinear model.

8. Fig. 10: The authors should represent these data as mean ± standard deviation of replications.

9. Fig. 12 should be incorporated to a Supplementary material.

10. Fig. 13: The resolution should also be strongly improved. It’s hard to read.

Author Response

  1. Title: “Myxococcus” should always be italiczed. The same for Escherichia coli (line 12). I recommend correcting /revise all manuscript.

Response: We have checked and revised the “Myxococcus” and “Escherichia coli” in manuscript.

  1. How is this system different to other reports to merit publication? The novelty of this study should be highlighted in the Introduction section.

Response: We have added the highlight of the study in the introduction section.

In the preliminary stages of this project, a strain of Myxococcus sp. V11 was isolated [https://doi.org/10.1016/j.pep.2021.105865]. Subsequent genome sequencing revealed the presence of a coding sequence (MCK8502360.1) for a starch debranching enzyme (MIsA). In this research, MIsA was heterologous expressed and characterized.

  1. The end of the introduction should be a remark of the interest of the study.

Response: We have added the interest of the study in end of the introduction.

  1. Line 143: The authors should express units for amylopectin. I recommend revise all manuscript.

Response: The unit of the amylopectin was 0.5% (w/v). We have added it to the manuscript.

  1. Lines 151/166: The authors should report the ionic strenght used in all buffer solutions.

Response: Ionic strength of 50 mM, we have added in the manuscript.

  1. Fig. 3 and Tables 1, 2, and 3: What is 100% relative activity. I recommend adding values for activity values at 100% in each Figure legend or Table.

Response: The enzyme activity at optimal reaction temperatures and pH levels was utilized as the control and defined as 100%. We have added it to the manuscript.

  1. Fig. 4: Linearization promotes the incorrect interpretation of models due to oversimplification. The authors should perform kinetic studies by fitting the nonlinear Michaelis-Menten model to the experimental data to determine apparent kinetic constant values. I recommend this study that confirms the superiority of parameter estimation (Km and Vmax) by nonlinear method over classical linearized model - https://doi.org/10.12793/tcp.2018.26.1.39 and https://doi.org/10.1021/acs.jchemed.6b00629. Therefore, the authors should calculate such apparent kinetic parameters using a nonlinear model.

Response: Thank you for your suggestion, I have referred to these two papers and made changes to the manuscript. We redrew the graph using nonlinear methods and recalibrated the Km and Vmax.

  1. Fig. 10: The authors should represent these data as mean ± standard deviation of replications.

Response: Three sets of replicates were done for each data set, and we have made the changes to the manuscript.

  1. Fig. 12 should be incorporated to a Supplementary material.

Response: We have added Figure 12 to the Supplementary Material and removed it from the manuscript.

  1. Fig. 13: The resolution should also be strongly improved. It’s hard to read.

Response: We redrew and placed in the supplementary material.

Round 2

Reviewer 1 Report

Comments and Suggestions for Authors

Revised version is satisfactory.

Comments on the Quality of English Language

Seems ok.

Reviewer 2 Report

Comments and Suggestions for Authors

The authors corrected the manuscript, s suggested. However, I recommend some minor corrections before publication

- Lines 91/98: This statement needs to be referenced.

- Line 95: "The sinergestic..."